# Expression Pattern of Tenascin-C, Matrilin-2, and Aggrecan in Diseases Affecting the Corneal Endothelium

**DOI:** 10.3390/jcm11205991

**Published:** 2022-10-11

**Authors:** Gréta Varkoly, Tibor G. Hortobágyi, Enikő Gebri, János Bencze, Tibor Hortobágyi, László Módis

**Affiliations:** 1Department of Ophthalmology, Szabolcs-Szatmár-Bereg County Hospitals, 4400 Nyíregyháza, Hungary; 2Albert Szent-Györgyi Medical School, University of Szeged, 6720 Szeged, Hungary; 3Department of Dentoalveolar Surgery and Dental Outpatient Care, Faculty of Dentistry, University of Debrecen, 4032 Debrecen, Hungary; 4Division of Radiology and Imaging Science, Department of Medical Imaging, Faculty of Medicine, University of Debrecen, 4032 Debrecen, Hungary; 5Department of Neurology, Faculty of Medicine, University of Debrecen, 4032 Debrecen, Hungary; 6Institute of Psychiatry Psychology and Neuroscience, King’s College London, London SE5 8AB, UK; 7Centre for Age-Related Medicine, Stavanger University Hospital, 4011 Stavanger, Norway; 8Institute of Neuropathology, University Hospital Zurich, 8091 Zurich, Switzerland; 9Department of Ophthalmology, Faculty of Medicine, University of Debrecen, 4032 Debrecen, Hungary

**Keywords:** aggrecan, cornea, extracellular matrix, Fuchs’ dystrophy, marilin-2, pseudophakic bullous keratopathy, tenascin-C

## Abstract

Purpose: The aim of this study was to examine the expression pattern of tenascin-C, matrilin-2, and aggrecan in irreversible corneal endothelial pathology such as pseudophakic bullous keratopathy (PBK) and Fuchs’ endothelial corneal dystrophy (FECD), which most frequently require corneal transplantation. Materials and methods: Histological specimens of corneal buttons removed during keratoplasty were investigated in PBK (*n* = 20) and FECD (*n* = 9) and compared to healthy control corneas (*n* = 10). The sections were studied by chromogenic immunohistochemistry (CHR-IHC) and submitted for evaluation by two investigators. Semiquantitative scoring (0 to 3+) was applied according to standardized methods at high magnification (400x). Each layer of the cornea was investigated; in addition, the stroma was subdivided into anterior, middle, and posterior parts for more precise analysis. In case of non-parametric distribution Mann–Whitney test was applied to compare two groups. Kruskal–Wallis and Dunn’s multiple comparisons tests have been applied for comparison of the chromogenic IHC signal intensity among corneal layers within the control and patient groups. Differences of *p* < 0.05 were considered as significant. Results: Significantly elevated tenascin-C immunopositivity was present in the epithelium and every layer of the stroma in both pathologic conditions as compared to normal controls. In addition, also significantly stronger matrilin-2 positivity was detected in the epithelium; however, weaker reaction was present in the endothelium in PBK cases. Minimal, but significantly elevated immunopositivity could be observed in the anterior and posterior stroma in the FECD group. Additionally, minimally, but significantly higher aggrecan immunoreaction was present in the anterior stroma in PBK and in the posterior stroma in both endothelial disorders. All three antibodies disclosed the strongest reaction in the posterior stroma either in PBK or in FECD cases. Conclusions: These extracellular matrix molecules disclosed up to moderate immunopositivity in the corneal layers in varying extents. Through their networking, bridging, and adhesive abilities these proteins are involved in corneal regeneration and tissue reorganization in endothelial dysfunction.

## 1. Introduction

The cornea is the first and most important refractive part of normal human eye. Composed of a six-layered avascular tissue, its curvature and transparency are essential for normal vision. 

The stroma is the thickest component of the cornea, covered by epithelium on the anterior and endothelium on the posterior surface. Its structure is built-up extracellular matrix (ECM) fibers and molecules; therefore, it is the most significant developmental challenge for transparency. Moreover, the transparency of the stroma depends on the well-organized microstructure of the major building elements such as collagen fibers, glycosaminoglycans (GAGs) and different types of proteoglycans (PGs). The most important collagens of the stroma are type I, V, and VI collagens. In addition, PGs composed of a protein core with covalently linked glycosaminoglycan side chains [1,2]. The major PGs of cornea are dermatan sulphate and keratan sulphate, while heparan sulphate, chondroitin sulphate and hyaluronan are present in a smaller amount [3].

The main significance of these PGs is the regulation of interfibrillar space characteristics and collagen fibril diameters [4]. PGs are also involved in adhesion, migration, signal transduction, proliferation of cells, fibril assembly, degradation, and inflammation [5]. Cells and molecules of the stroma form a distinct and strictly organized network of approximately 60 lamellae. Since visible light can pass through, this unique architecture is the key determinant of corneal transparency. The collagen fibrils taking part in the construction of this network are perpendicular to the corneal surface, uniform in size (24–25 mm) and have regular spacing between them [6]. For light rays to cross through the corneal stroma the undamaged structure of proteoglycans is also required [7].

In specific diseases, this structural and physiological status is disturbed, with tissue degeneration leading to irreversible stromal opacity. After a period, conservative treatment is ineffective and unsuccessful, and thus, corneal transplantation is necessary.

Previously, a wide range of cell–cell and cell–matrix interactions and adhesion molecules have been studied, and different collagen types investigated, as possible participants in the emergence of corneal opacity and pathologic alterations [8,9,10,11,12].

However, smaller ECM proteins families, such as tenascin-C, matrilin-2, and aggrecan have not yet been thoroughly investigated in corneal diseases. Tenascin-C is a hexameric ECM glycoprotein with immediately increased expression in injury and infection. At cellular level, it is responsible for various dynamic cellular activities, such as cell adhesion, cell–ECM interaction, tissue remodeling, and proliferation [13,14,15,16,17].

Matrilin-2 is a member of the von Willebrand factor type A-like module superfamily [18]. The matrilins form a family of four, with each member binding to different types of collagenous and non-collagenous ECM structures, determining tissue stability via various interactions in the ECM [19]. Matrilin-2 has yet only been thoroughly studied in cartilage, muscle, and nerve, even though it is a crucial component of basement membranes throughout the body, including the corneal epithelial basement membrane [18,19,20]. Matrilins mostly mediate interactions between collagen-containing fibrils and other matrix constituents, such as aggrecan.

Aggrecan consists of a core protein and glycosaminoglycan side chains. The core protein is composed of three globular domains (G1, G2, and G3) and two interglobular regions, the latter of which is a large extended region between G2 and G3 for glycosaminoglycan chain attachment. Aggrecan plays a key role in forming a properly hydrated ECM structure, and it has been extensively studied in normal and osteoarthritic cartilage [21,22]. In addition, with other PGs and GAGs, aggrecan has a certain capacity in wound healing and fibrosis [23].

The aim of this study was to examine the expression of tenascin-C, matrilin-2, and aggrecan in irreversible corneal endothelial pathology such as pseudophakic bullous keratopathy (PBK) and Fuchs’ endothelial corneal dystrophy (FECD), which most frequently require corneal transplantation. 

## 2. Materials and Methods

### 2.1. Corneal Specimens

Corneal buttons were collected from patients who underwent corneal transplantation at the Department of Ophthalmology, University of Debrecen, Hungary. In 20 cases, the diagnosis was PBK with the mean age of 71.7 ± 7.55 years (Table 1), and in 9 cases, the indication for surgery was FECD with the mean age of 68.44 ± 11.35 years (Table 2). 

Ten healthy age-matched corneas served as normal controls (from 3 females and 2 males with the mean age 63.73 ± 7.3 years) harvested from cadavers at the Department of Pathology, University of Debrecen. Control corneas had negative history of previous ocular inflammation and surgical intervention. Those corneas used were ones that would have been suitable for transplantation during donor collection, but were isolated for scientific purposes. Thus, after slit-lamp examination, they were not preserved but immediately transferred to fixative.

The study was approved by the Institutional Scientific and Research Ethics Board (4468-2015). 

### 2.2. Preparation of Histological Sections

After harvesting, corneas were immersion-fixed using a formaldehyde solution (pH = 7.4; 10% *v*/*v*) changed subsequently to paraformaldehyde (pH = 7.4; 4% *v*/*v*) overnight and then embedded into paraffin wax according to standard laboratory procedures. Then, 7 µm thick sections were cut from the paraffin embedded blocks (Leica RM2245 microtome, Leica Biosystems Nussloch Gmbh, Nussloch, Germany), coverslipped with DPX (BDH Laboratory Supplies, Poole, UK) and left to dry overnight at room temperature.

### 2.3. Immunohistochemistry (IHC) and Chromogenic Immunohistochemistry (CHR-IHC)

For the IHC procedures, the following antibodies were used: tenascin-C mouse monoclonal antibody (DB7; ab86182, Abcam, Cambridge, UK) at 1:50 dilution; matrilin-2 rabbit polyclonal antibody (ARP57667_P050, Aviva Systems Biology, Corp., San Diego, CA, USA) at 1:300 dilution; aggrecan rabbit polyclonal antibody (MerkMillipore, AB1031, Merck Life Science Kft., an affiliate of Merck KGaA, Darmstadt, Germany) at 1:750 concentration was used with a common standard overnight immunohistochemistry protocol [9].

After de-waxing and rehydration according to standard protocols, the endogenous peroxidase activity was blocked by Biocare Medical Peroxidazed 1 blocking reagent (Biocare Medical, Pacheco, CA, USA) in Biocare Medical Peroxidazed diluent (Biocare Medical) diluted at 1:3, following the manufacturer’s instructions [24]. Afterwards, heat-induced epitope retrieval was performed in a microwave oven (5 min at 800 W, 2 × 5 min at 250 W) with sodium citrate buffer (pH = 6.0) as buffer solution, in case of tenascin-C and matrilin-2. Aggrecan epitope retrieval was performed with chondroitinase ABC in sodium citrate buffer (pH = 8.0) (37 °C, 1 h). 

The non-specific IHC reaction was blocked by 1% (*v*/*v*) bovine serum albumin (BSA) (Sigma-Aldrich, St. Louis, MO, USA) for one hour, at room temperature. 

Antibody dilution was 1:50 for tenascin-C, 1:300 for matrilin-2, and 1:750 for aggrecan. Preliminary experiments were performed to establish optimal antibody concentration and incubation time. As a result, tenascin-C- and aggrecan-treated sections were incubated overnight, at 4 °C, while for matrilin-2, the incubation time was 45 min, at room temperature. After washing out the primary antibody with Tris-buffered saline (TBS) (Sigma-Aldrich, TRIS-Buffered Saline 20× solution, 20 mmol Tris, pH 7.4), biotin-free secondary antibodies were added (MACH 4 Universal HRP-Polymer, Biocare Medical) for 30 min, at room temperature, to aggrecan and matrilin slides. We used Mouse Probe to tenascin-C slides, and these were incubated for 10 min, at room temperature, then HRP-Polymer was added. The incubation time was 20 min. After washing in TBS an immunoperoxidase technique using 3,3′-diaminobenzidine tetrahydrochloride (DAB) (Biocare Medical) was applied as chromogen in humid chambers, at room temperature. Then, the specimens were rinsed in running tap water for 10 min and then in distilled water for one minute. 

Subsequently, sections were treated with Harris’ hematoxylin as a nuclear counterstain. This step was followed by routine dehydration in ascending concentrations of ethyl alcohol solutions. Finally, the corneal buttons were covered with DPX mounting medium (Sigma-Aldrich, St. Louis, MO, USA) and were coverslipped manually. 

We performed negative controls, omitting the primary antibody.

### 2.4. Evaluation and CHR-IHC Scoring

Semiquantitative image analysis was performed by two authors of this article (GV, TH). A Nikon eclipse 80i light microscope was used at the magnification of 400x, and analysis was performed according to standardized methods of CHR-IHC assessment as described earlier [25]. The semiquantitative scoring system had 5 scales (0; 0.5+; 1+; 2+; 3+), where 0 means no staining (negative), 0.5+ minimal, 1+ mild, 2+ moderate and 3+ marked positivity. Every layer of the cornea was analyzed. In the epithelium and endothelium 100 cells were counted in each section and calculated the arithmetic mean of the scores. In the other three layers (Bowman’s membrane, stroma, Descemet’s membrane); however, the general CHR sign intensity was established throughout the thickness of that particular layer. For precise evaluation, the stroma was divided into three parts, anterior, middle, and posterior subdivision, respectively. 

### 2.5. Statistical Analysis

Statistical analysis was performed using GraphPad Prism 8.0. Variability of values is given as the standard error of the mean (SEM). Results are reported as mean ± SEM. Data distributions were tested for normality using the Shapiro–Wilk normality test. In case of non-parametric distribution Mann–Whitney test was applied to compare two groups (PBK vs. control, FECD vs. control, FECD vs. PBK) in each examined layer (epithelium, Bowman membrane, anterior stroma, middle stroma, posterior stroma, Descemet, endothelium) regarding the three assessed proteins (tenascin-C, matrilin-2, aggrecan). Kruskal–Wallis test followed Dunn’s multiple comparison post hoc non-parametric test have been applied for comparison of the chromogenic IHC signal intensity among corneal layers within the control and patient groups (Appendix A). Differences of *p* < 0.05 were considered as significant.

## 3. Results

### 3.1. Tenascin-C CHR-IHC

The distribution of tenascin-C immunopositivity is summarized in Table 3 and demonstrated in Figure 1. 

In the surgically removed pathologic PBK and FECD groups, the epithelium disclosed moderate immunoreactivity, and it was significantly higher in PBK specimens as compared to controls. Bowman’s membrane and stoma were also stained more intensely than normal corneas and the difference was significant in all stromal layers. Both in PBK and in FECD, the posterior stroma had the highest tenascin-c expression (1.85 and 1.67, respectively). Descemet’s membrane had minimal immunopositivity in PBK and controls, and no staining was present in FECD corneas. There was mild but not remarkable difference in the immuno-expression pattern in the in endothelium in the three studied groups.

Comparing PBK and FECD cases with normal controls and, moreover, diseased groups with each other, the following statistics were obtained (Table 4). 

Comparing PBK and FECD groups only the anterior stroma showed statistically significant difference in immunolabelling intensity in PBK cases. 

We compared tenascin-C expression in corneal layers within all investigated groups. Data are shown in Appendix A.

### 3.2. Matrilin-2 CHR-IHC

The expression of matrilin-2 immunopositivity is summarized in a table and a figure (Table 5 and Figure 2).

The surgical specimens showed moderate staining in the epithelium; moreover, in the PBK group, the difference was significant compared to controls. The acellular layers (Bowman’s and Descemet’s membrane) and all stromal parts disclosed minimal or no immunoreaction. However, these corneal components in FECD corneas were still more heavily stained than healthy cases; in addition, the anterior and posterior part of the stroma and Descemet’s membrane showed significant immunopositivity. In PBK, the endothelium had significantly decreased immunostaining (0.70) as compared to controls (1.70), while in FECD, there was no difference.

Comparing PBK and FECD specimens, almost all layers showed significant difference, except Bowman’s membrane and posterior stroma (Table 6). In PBK cases, the epithelium stained moderately and significantly stronger compared to FECD group. In contrast, in the FECD group, the anterior and middle stroma, Descemet’s membrane and endothelium disclosed significantly higher immunoreactivity than PBK corneas. 

Statistical analysis of the corneal layers within controls, PBK and FECD groups, resulted in the following data, shown in Appendix A.

### 3.3. Aggrecan CHR-IHC

The pattern of aggrecan expression is summarized in Table 7 and Figure 3.

In PBK and FECD, the epithelium and Bowman’s membrane disclosed mild immunostaining, but the difference was not significantly different compared to controls. In both pathological groups, corneal stroma exhibited minimal labelling. Anterior stroma in PBK and posterior stroma in both diseased groups differed significantly from normal cases. The Descemet’s membrane did not stain. Corneal endothelium had mild immuno-expression, in all groups.

Comparing PBK and FECD cases, there was no evident difference in aggrecan expression (Table 8).

Aggrecan expression showed the following differences between corneal layers within the investigated groups, shown in Appendix A.

Cumulative statistical data were also calculated to provide a comprehensive insight about the significance of the investigated molecules (Table 9). 

The immunoreactivity of tenascin-C was significantly elevated in diseased corneas as compared to normal cases; moreover, PBK showed significantly stronger positivity than FECD cases.

Matrilin-2 disclosed significantly higher immunopositivity in comparing FECD to normal controls and even to PBK.

In comparison of PBK to FECD cases, there were no relevant differences regarding the expression pattern of aggrecan, while significantly stronger immunoreactivity was present in PBK and FECD cases as compared to normal corneas.

Demonstrating the results, the prominent finding was the significantly elevated expression of tenascin-C in the epithelium in PBK, and in every layer of the stroma in both pathologic conditions (Figure 4). 

Significantly stronger matrilin-2 positivity was detected in the epithelium, and decreased reaction was present in the endothelium in PBK cases. Minimal, but significantly elevated immunopositivity could be observed in the anterior and posterior stroma and, furthermore, in Descemet’s membrane, which was unique in the FECD group (Figure 5). 

Mild-to-moderate aggrecan immunoreaction was present in the corneal stroma in both endothelial disorders. Aggrecan disclosed the strongest reaction in the posterior stroma either in PBK or in FECD cases (Figure 6).

The average distribution of tenascin-C, matrilin-2 and aggrecan immunopositivity in PBK, FECD and normal controls is demonstrated on a schematic figure (Figure 7).

## 4. Discussion

The present study investigated the expression pattern of three different extracellular matrix proteins in corneal endothelial pathologies such as pseudophakic bullous keratopathy and Fuchs’ endothelial corneal dystrophy, in which the final solution is corneal transplantation.

The most conspicuous finding of the study was the significantly elevated immunoreactivity for tenascin-C in the corneal epithelium in PBK and each layer of the stroma in both disorders as compared to normal samples. Significantly elevated matrilin-2 positivity were seen in the epithelium in PBK, contrariwise in the endothelium, where significantly lower immunoreaction was detected. In FECD, the anterior and posterior part of the stroma and Descemet’s membrane were stained significantly uniquely in FECD. In addition, matrilin-2 expression showed the most decided differences in diseased groups. Aggrecan expressed higher immunopositivity in the anterior part of the stroma in PBK and FECD also, and in the posterior stroma in PBK. There were no significant differences between diseased groups. 

The other striking difference was that the anterior and posterior part of the stroma showed significantly stronger positivity with all three investigated proteins in both diseases (except in PBK with matrilin-2 and anterior stroma with aggrecan in FECD). Moreover, all three antibodies disclosed the strongest reaction in the posterior stroma either in PBK or in FECD cases. 

Tenascin-C, as an adhesion-modulating extracellular matrix protein, binds to numerous ECM components, cell membrane elements, soluble factors and to fibronectin [13,14,15,16,17]. This latter interaction is responsible for cellular upregulation of matrix metalloproteinase expression, one of the important factors in tissue degradation in wound healing [26]. Moreover, tenascin-C is capable to regulate mechanical and adhesion interactions in cell–cell and cell–ECM relations with impact on intracellular signal transduction pathways. Specific conditions induce persistent tenascin-C production, such as infection, inflammation, wound-healing processes, cancer, autoimmune and fibrotic diseases [13,14].

Regarding the cornea, earlier publications reported higher tenascin-C level in bullous keratopathy, keratoconus, and its role has been demonstrated in corneal inflammation, fibrosis, scarring, neovascularization, and wound healing [27,28,29,30]. Previously, our working group observed that both tenascin-c and matrilin-2 proteins are unequivocally expressed resolutely in lattice type I and granular stromal dystrophy [31]. Prior reports have described excessive tenascin-C expression in the epithelium, basement membrane, subepithelial layer, and posterior stroma in bullous keratopathy [27,32]. This was mostly consistent with our observations. Moreover, we found similar distribution pattern in FECD. 

The common characteristic clinical finding in PBK and FECD is corneal endothelial cell destruction and loss. This may serve as a possible explanation for the mild (but elevated) tenascin-C immunopositivity in this corneal layer. The subsequent stromal edema usually progresses to the epithelium, causing severe pain. Finally, stromal degeneration and opacification develops, causing visual disturbance. 

Matrilin-2, as a multiadhesion basement membrane component, interacts with fibrillins, integrins, and other ECM proteins [33]. The function of matrilin-2 is highly variable in the human body, such as promotion of axonal growth and Schwann cell migration during peripheral nerve regeneration, modulation of dermal wound healing, tumor development, and muscle regeneration [33,34,35,36].

However, only a few reports have been studied the role of matrilin-2 in human corneas. Earlier studies disclosed the presence of matrilin-2 in basements membranes [37]. Previously we demonstrated elevated matrilin-2 levels in the corneal epithelium and stroma in different corneal stromal dystrophies [31].

In the present study, while we observed significant immunoreaction in the epithelium in PBK, in contrast, there was elevated staining in the anterior, posterior stroma and Descemet’s membrane in FECD. Significant Descemet’s membrane positivity was unique in Fuchs’ endothelial dystrophy. Matrilin-2′s role in wound healing has been described previously. Although wound healing processes take place in both investigated disorders, matrilin-2 expression pattern showed differences between PBK and FECD. This molecule is able to make several connections with ECM components, such as collagens, fibrillins, laminin and fibronectin, and in addition, fibroblasts produce matrilin-2 [33]. The most interesting finding was the significantly stronger positivity in endothelium and Descemet’s membrane in FECD as compared to PBK. We can hypothesize that matrilin-2 may bind transforming growth factor beta-induced protein (TGF-β), a component of guttae, characteristic to FECD. Jurkunas et al. published that TGF-β levels are elevated in the Descemet membrane and endothelium of FECD corneas [38]. In addition, another study may confirm our hypothesis. Szalai et al. demonstrated elevated matrilin-2 expression in stromal deposits in granular type I and lattice corneal dystrophy. These diseases are caused by TGFBIp mutations [31]. The more pronounced positivity in Descemet’s membrane and endothelium can be attributed to the extended accumulation of fibronectin in guttae, which makes connections with matrilin-2. 

PBK develops after cataract surgery. Several preoperative (older age, lower endothelial cell count prior surgery) and intraoperative factors (excessive manipulation and use of phacoemulsification energy, toxic irrigating solutions, and posterior capsular rupture with vitreous loss) contribute to its progression. In FECD, excrescences of Descemet’s membrane with abnormal posterior stromal layer are present, leading to corneal guttae and endothelial cell loss. However, both in PBK and FECD, the key feature in the pathogenesis is endothelial cell loss, which causes clinically endothelial decompensation, consequential stromal edema, epithelial bullae with recurrent pain and decreased vision, although FECD has slower progression. The more pronounced presence of matrilin-2 in the stromal layers in FECD may explain the indolent course of this disease. 

Aggrecan, a proteoglycan core protein, consists of three globular domains, G1, G2, and G3, and they have a certain role in aggregation, cell adhesion, and hyaluronic acid binding. The latter is responsible for providing a necessary ECM structure and function by maintaining the specifically hydrated gel structure [21]. Aggrecan also acts as a binding molecule in the ECM, bridging various matrix components and cells. The structure of aggrecan changes over the period of life due to synthetic and degradative processes [22].

The aggrecan molecule has hardly been studied before in the human cornea. Increased aggrecan production has been reported in the sclera of myopic chicks [39]. Another paper disclosed increased aggrecan immunolabeling in sclerocornea compared to normal control [40]. In sclerocornea, diffuse strong positivity was present through the whole specimen. In addition, in healthy cases, the authors found marked positivity in the epithelium and mild-to-moderate immunostaining in the stroma.

In the present study, we found elevated aggrecan expression in stromal layers. In PBK anterior and posterior stroma, in FECD, only posterior stroma showed significant aggrecan positivity. In both disease, elevated TGF-β levels stimulate fibroblasts to aggrecan expression [41], a molecule which makes interactions with tenascin-C [13]. Elevated concentration of aggrecan induces high osmotic gradient, via binding Na^+^ ions to its GAG side chains [21], aggravating the edema resulting from endothelial cell loss and Na^+^/K^+^ ATP-ase defect. Comparing the diseased groups with each other, there were no significant differences in aggrecan expression, but in anterior and middle stroma, milder positivity was detectable in FECD cases. Matrilin-2 may silence the TGF-β mediated pathways and, in this way, aggrecan expression of myofibroblasts. 

Based our findings, there is clinical and pathological relevance of our study. Initial endothelial cell damage with a concomitant corneal edema leading to increased intrastromal pressure is a common feature in the pathomechanism of PBK and FECD. Excessive stromal fluid accumulation exposes keratocytes to mechanical stress. As a result, stromal keratocytes can contract and stretch the surrounding collagen fibers [27]. If the endothelial pathology progresses, epithelial bullae develop with basement membrane injury. This process triggers further structural and functional changes with epithelium-derived cytokine invasion to the stroma and initiate differentiation of myofibroblast precursor cells [20,37]. In addition, accumulation of matrix metalloproteinase and inflammatory cells takes place [42]. After repair, keratocytes undergo apoptosis and tissue remodelling [37,42]. With the adhesive and lubricant function of tenascin-c and aggrecan, and with the modulating properties of ECM assembly and matrix–cell communication of matrilin-2, these proteins seem to be involved in the regeneration and reorganization of the corneal ECM in these degenerative processes. The presence and distribution of the studied molecules and the hypothesized bindings and effects are summarized and demonstrated schematically in Figure 8 and Figure 9.

Some patients suffered from systemic diseases, such as NIDDM or RA in PBK and FECD cases, also seen in Table 1 and Table 2. These disorders can enhance the fibrotic changes in the cornea.

In these specific endothelial diseases with corneal opacification, the gold standard treatment is corneal transplantation, and posterior lamellar grafting especially is becoming more common. However, the study of these proteins and a precise understanding of their expression may help us to develop new conservative treatment methods. These could include the development of eye drops containing matrilin-2, fibronectin, insulin-like growth factor-1, substance-P peptide for corneal damage, or blocking extracellular high-mobility group box 1 with glycyrrhizin [43,44]. 

Furthermore, gene therapies could maintain the appropriate balance of tenascin-C, matrilin-2 and aggrecan may delay the progression of the diseases. Tenascin-C-specific nanobodies may be useful to inhibit the immune-suppressive function and other functions of tenascin-C [45]. BMP7 and HGF gene therapy treats corneal fibrosis and reduces the level of alpha smoot muscle actin via triggering the Smad 1/5/8 signaling pathway [46]. Moreover, PPARP agonist reduces the TGF-β-induced aggrecan synthesis, which may also reduce corneal opacities [47].

In summary, the present study investigated the immunohistochemical expression pattern of tenascin-C, matrilin-2, and aggrecan in advanced forms of corneal endothelial pathology such as pseudophakic bullous keratopathy and Fuchs’ endothelial corneal dystrophy. These extracellular matrix molecules disclosed mild-to-moderate immunopositivity in the corneal layers in varying extents. Through their bridging and connective abilities, these proteins are involved in corneal wound healing and regeneration.

## Figures and Tables

**Figure 1 jcm-11-05991-f001:**
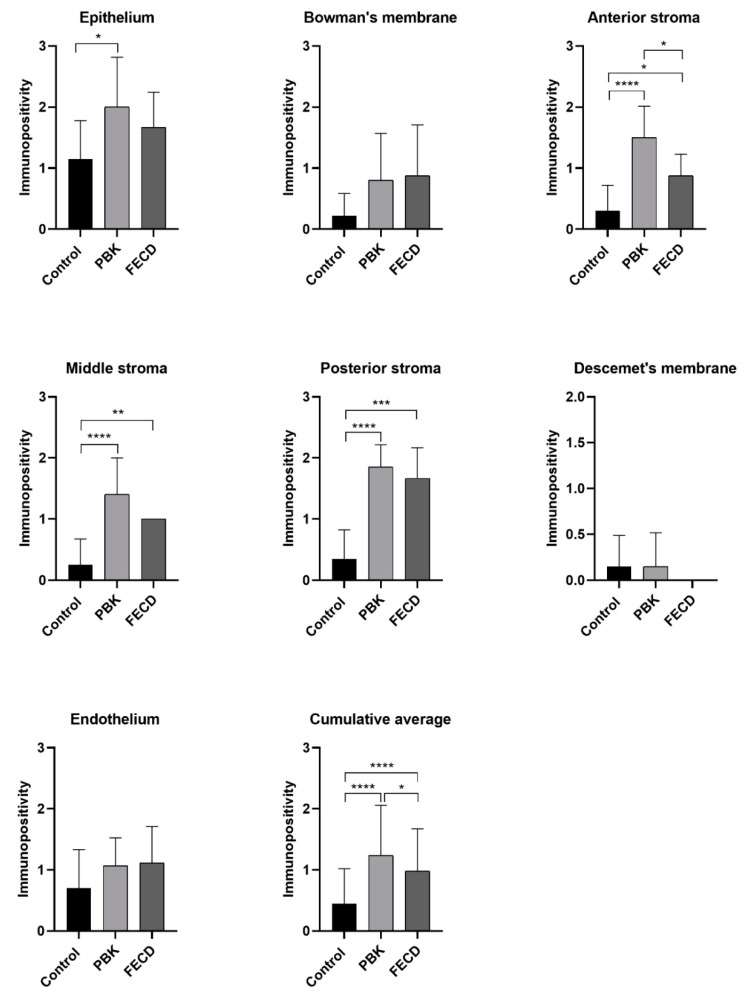
The average tenascin-C immunopositivity in the different corneal layers in normal and pathologic conditions. PBK = pseudophakic bullous keratopathy, FECD = Fuchs’ endothelial corneal dystrophy. Columns indicate the average immunopositivity (0 means no staining, 0.5+ minimal, 1+ mild, 2+ moderate, 3+ marked positivity). * = level of significance (*p* ˂ 0.05 *; *p* ˂ 0.01 **; *p* ˂ 0.001 ***; *p* ˂ 0.0001 ****).

**Figure 2 jcm-11-05991-f002:**
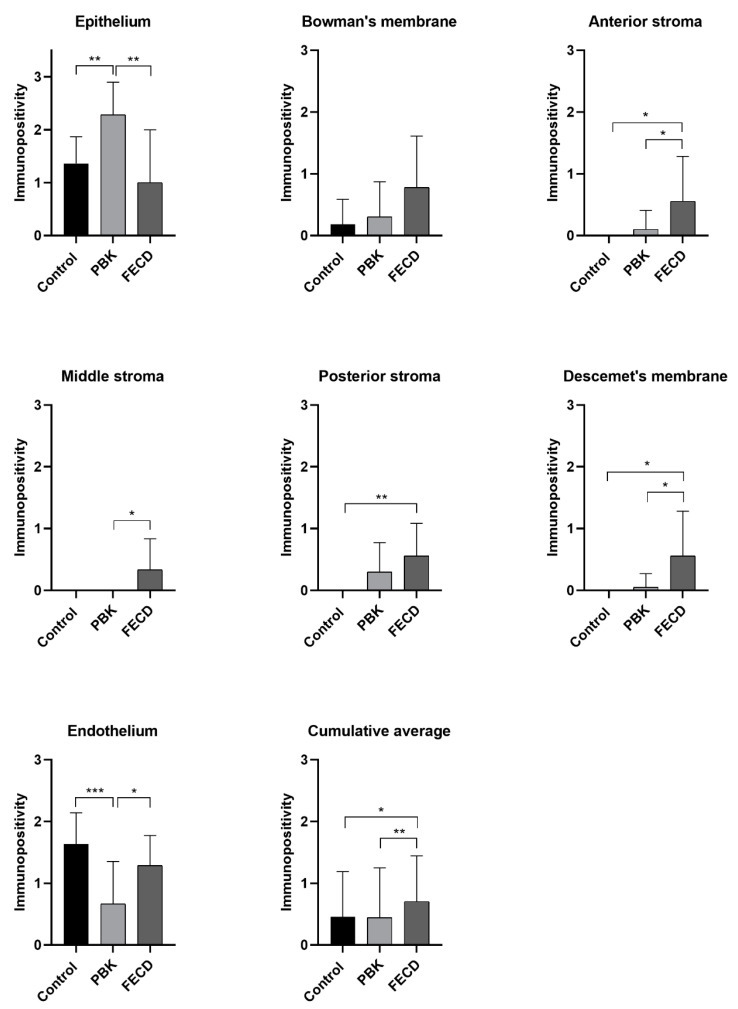
The average matrilin-2 immunopositivity in the different corneal layers in normal and pathologic conditions. PBK = pseudophakic bullous keratopathy, FECD = Fuchs’ endothelial corneal dystrophy. Columns indicate the average immunopositivity (0 means no staining, 0.5+ minimal, 1+ mild, 2+ moderate, 3+ marked positivity). * = level of significance (*p* ˂ 0.05 *; *p* ˂ 0.01 **; *p* ˂ 0.001 ***).

**Figure 3 jcm-11-05991-f003:**
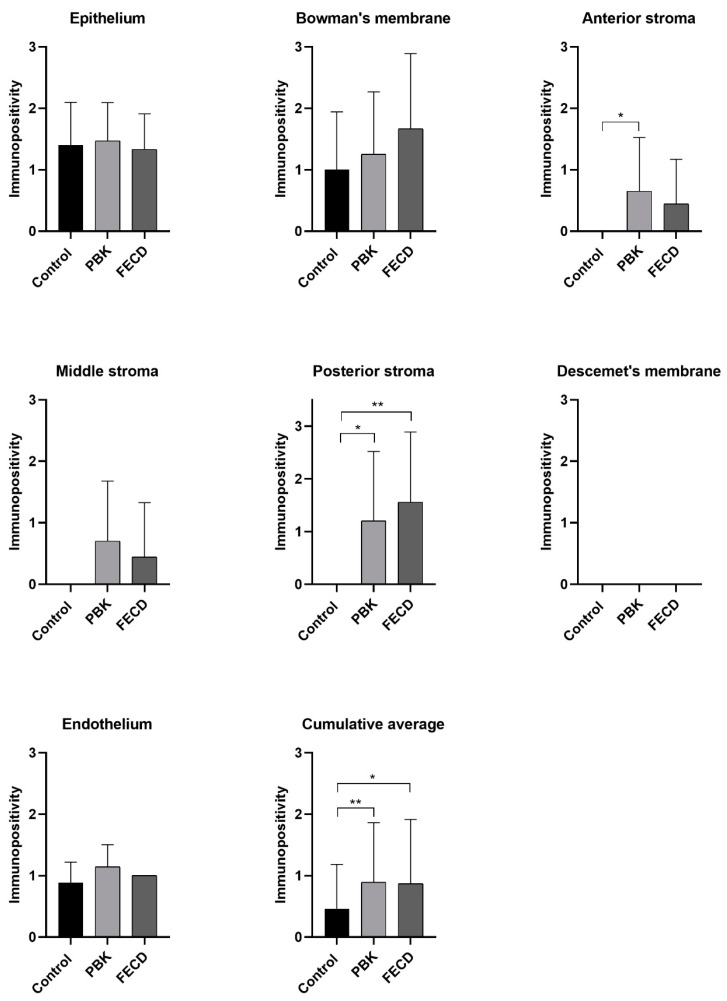
The average aggrecan immunopositivity in the different corneal layers in normal and pathologic conditions. PBK = pseudophakic bullous keratopathy, FECD = Fuchs’ endothelial corneal dystrophy. Columns indicate the average immunopositivity (0 means no staining, 0.5+ minimal, 1+ mild, 2+ moderate, 3+ marked positivity). * = level of significance (*p* ˂ 0.05 *; *p* ˂ 0.01 **).

**Figure 4 jcm-11-05991-f004:**
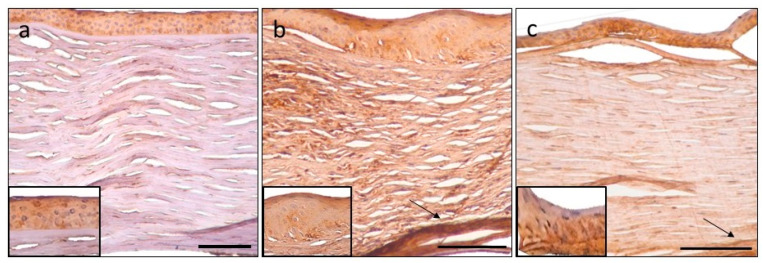
Tenascin-C immunohistochemistry. Control cornea with moderate positivity in the epithelium (**a**). Intense expression in PBK in the epithelium and throughout the stroma, most prominent in the pre-Descemet area (arrow) (**b**). Moderate epithelial and mild stromal staining in FECD, stromal distribution pattern is similar to PBK, with prominent posterior stromal labelling (arrow) (**c**). Epithelial signal intensity (insets). Scale bar indicates 100 μm.

**Figure 5 jcm-11-05991-f005:**
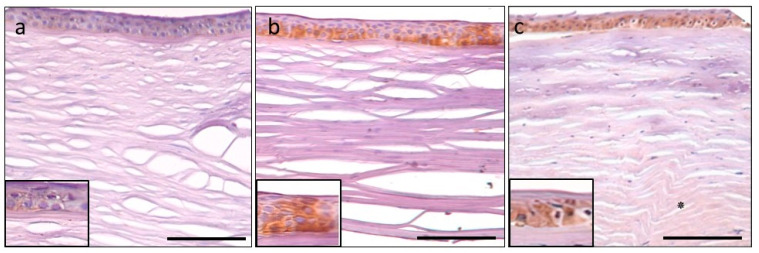
Matrilin-2 immunohistochemistry. Control cornea with minimal immunopositivity in the epithelium and endothelium and unstained stroma (**a**). Moderate epithelial immuno-expression in PBK and the rest of the tissue is unstained (**b**). FECD specimen also disclosed moderate epithelial staining and minimal stromal positivity is present in the posterior area (asterisk) (**c**). Epithelial signal intensity (insets). Scale bar refers to 100 μm.

**Figure 6 jcm-11-05991-f006:**
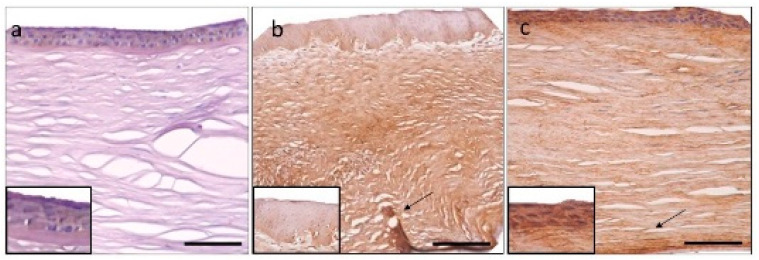
Aggrecan immunohistochemistry. Control section with minimal immunopositivity in the epithelium (**a**). Mild-to-moderate epithelial and stromal immunopositivity in PBK (**b**) and FECD (**c**) is present. Posterior stroma disclosed more intense labelling (arrow). Epithelial signal intensity (insets). Scale bar indicates 100 μm.

**Figure 7 jcm-11-05991-f007:**
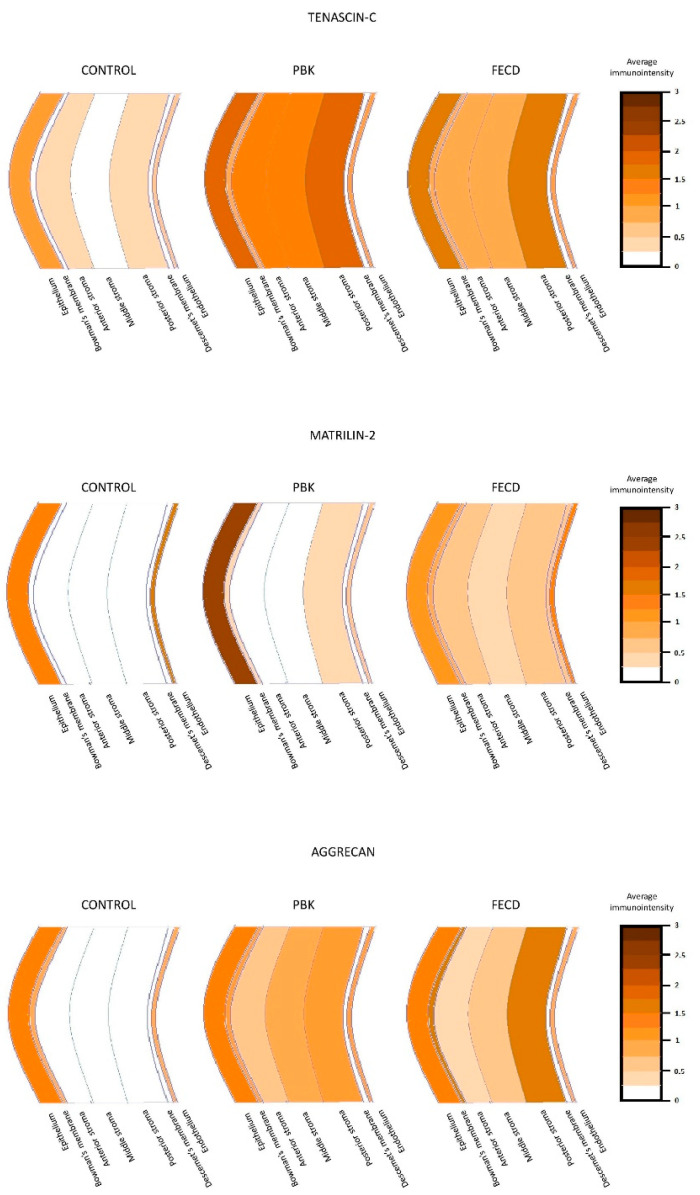
The average immunointensity of tenascin-C, matrilin-2 and aggrecan in normal controls, PBK and FECD groups. PBK = pseudophakic bullous keratopathy, FECD = Fuchs’ endothelial corneal dystrophy. Scale bar and colour mark indicates the average immunointensity of the investigated ECM proteins.

**Figure 8 jcm-11-05991-f008:**
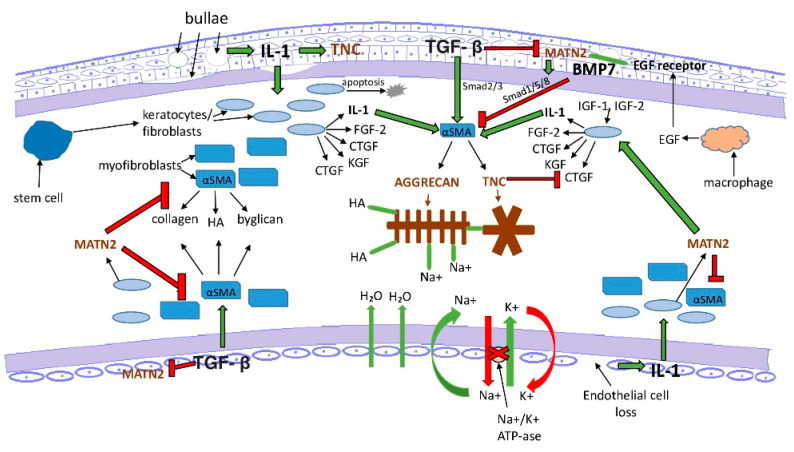
Proposed wound-healing mechanisms in PBK. The presence, distribution and the hypothesized bindings and effects of tenascin-C, matrilin-2 and aggrecan. αSMA = alpha smooth muscle actin, BMP7 = bone morphogenic factor 7, CTGF = connective tissue growth factor, EGF = epidermal growth factor, FGF-2= fibroblast growth factor 2, HA = hyaluron acid, IGF-1 = insulin-like growth factor 1, IGF-2 = insulin-like growth factor 2, IL-1 = interleukin-1, KGF = keratocyte growth factor, MATN2 = matrilin-2, TGF-β = transforming growth factor beta, and TNC = tenascin-C. Green arrows with black contour = positive trigger, red arrows with black contour = inhibition, and green line = bounding.

**Figure 9 jcm-11-05991-f009:**
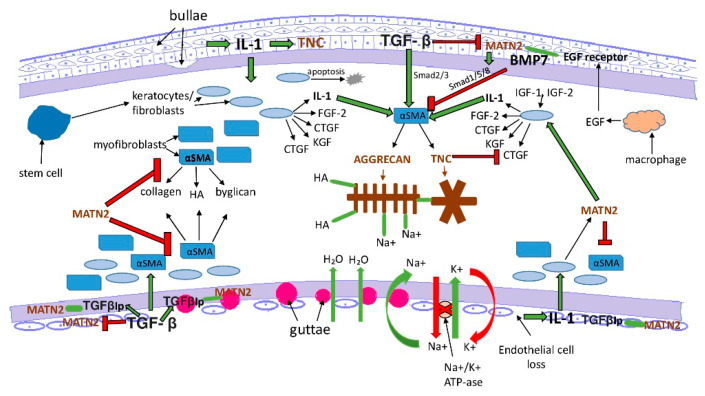
Proposed wound-healing mechanisms in FECD. The presence, distribution and the hypothesized bindings and effects of tenascin-C, matrilin-2 and aggrecan. αSMA = alpha smooth muscle actin, BMP7 = bone morphogenic factor 7, CTGF = connective tissue growth factor, EGF = epidermal growth factor, FGF-2 = fibroblast growth factor 2, HA = hyaluron acid, IGF-1 = insulin-like growth factor 1, IGF-2 = insulin-like growth factor 2, IL-1 = interleukin-1, KGF = keratocyte growth factor, MATN2 = matrilin-2, TGF-β = transforming growth factor beta, and TNC = tenascin-C. Green arrows with black contour = positive trigger, red arrows with black contour = inhibition, and green line = bounding.

**Table 1 jcm-11-05991-t001:** Clinical characteristics of PBK patients.

Patient	Age	Sex	Clinical Findings	Systemic Disease
1	84	female	stromal decompensation, AC IOL	IDDM
2	67	female	stromal decompensation, PC IOL, glaucoma	NIDDM
3	75	female	stromal decompensation, glaucoma	-
4	79	female	stromal decompensation, aphakia	ischaemic heart disease, hypertension
5	84	female	stromal opacity, AC IOL	NIDDM
6	71	female	stromal decompensation, PC IOL, myopia	-
7	74	female	stromal decompensation, aphakia	thrombocythaemia
8	73	female	stromal decompensation, PC IOL	hypertension
9	68	female	stromal decompensation, PC IOL, glaucoma, macular degeneration	Sjögren syndrome
10	66	female	stromal decompensation, aphakia, glaucoma	hypertension
11	77	male	stromal decompensation, PC IOL, macular degeneration	-
12	64	female	stromal opacity	-
13	79	female	stromal decompensation, glaucoma, PC IOL	-
14	63	male	stromal decompensation, PC IOL	RA, cardiomyopathy, hypertension
15	82	female	stromal decompensation, PC IOL, glaucoma	hypertension
16	67	male	stromal opacity, PC IOL, glaucoma	-
17	68	female	stromal decompensation, PC IOL	-
18	70	male	stromal decompensation. AC IOL, glaucoma	-
19	68	male	stromal decompensation, PC IOL, myopia	-
20	55	male	stromal opacity, PC IOL, glaucoma	-

IDDM = insulin-dependent diabetes mellitus, PBK = pseudophakic bullous keratopathy, NIDDM = non-insulin-dependent diabetes mellitus, AC = anterior chamber, PC = posterior chamber, IOL = intraocular lens, RA = rheumatoid arthritis.

**Table 2 jcm-11-05991-t002:** Clinical characteristics of FECD patients.

Patient	Age	Sex	Clinical Findings	Systemic Disease
1	70	female	stromal opacity, shallow AC	RA
2	70	female	stromal opacity, glaucoma	-
3	70	female	stromal edema, PC IOL, glaucoma	NIDDM, hypertension
4	66	female	stromal edema, glaucoma, PC IOL	-
5	80	female	stromal opacity, PC IOL, amblyopia	NIDDM
6	62	female	stromal opacity, glaucoma, amblyopia	RA, hypertension
7	84	female	stromal edema, cataract	hypertension
8	44	female	stromal edema, shallow AC	-
9	70	male	stomal opacity, PC IOL, glaucoma	hypertension

FECD = Fuchs’ endothelial corneal dystrophy, NIDDM = non-insulin-dependent diabetes mellitus, AC = anterior chamber, PC = posterior chamber, IOL = intraocular lens, RA = rheumatoid arthritis.

**Table 3 jcm-11-05991-t003:** Tenascin-C immunopositivity in the different corneal layers in normal and pathologic conditions.

	PBK	FECD	Control
Epithelium	2.00 * ±0.79	1.67 ±0.58	1.15 ±0.63
Bowman’s membrane	0.80 ±0.77	0.88 ±0.83	0.22 ±0.36
Anterior stroma	1.50 **** ±0.51	0.88 * ±0.35	0.30 ±0.42
Middle stroma	1.40 **** ±0.60	1.00 ** ±0.00	0.25 ±0.42
Posterior stroma	1.85 **** ±0.37	1.67 *** ±0.50	0.35 ±0.47
Descemet’s membrane	0.15 ±0.37	0.00	0.15 ±0.34
Endothelium	1.06 ±0.46	1.11 ±0.60	0.70 ±0.63

PBK = pseudophakic bullous keratopathy, FECD = Fuchs’ endothelial corneal dystrophy. Numbers indicate semiquantitative scoring ± SEM (0 means no staining, 0.5+ minimal, 1+ mild, 2+ moderate, 3+ marked positivity). * = level of significance (*p* ˂ 0.05 *; *p* ˂ 0.01 **; *p* ˂ 0.001 ***; *p* ˂ 0.0001 ****).

**Table 4 jcm-11-05991-t004:** Comparison of tenascin-C expression in the different corneal layers in normal and diseased groups.

	PBK vs. Control	FECD vs. Control	PBK vs. FECD
Epithelium	0.0105	0.3706	0.6945
Bowman’s membrane	0.0579	0.0883	0.9685
Anterior stroma	<0.0001	0.0109	0.0162
Middle stroma	<0.0001	0.0013	0.0622
Posterior stroma	<0.0001	0.0004	0.3391
Descemet’s membrane	0.9999	0.4771	0.5360
Endothelium	0.0713	0.1500	0.8661

PBK = pseudophakic bullous keratopathy, FECD = Fuchs’ endothelial corneal dystrophy. Numbers indicate adjusted *p* values, and red numbers are statistically significant data.

**Table 5 jcm-11-05991-t005:** Matrilin-2 immunopositivity in the different corneal layers in normal and pathologic conditions.

	PBK	FECD	Control
Epithelium	2.29 ** ±0.61	1.00 ±1.00	1.30 ±0.50
Bowman’s membrane	0.30 ±0.57	0.78 ±0.83	0.20 ±0.40
Anterior stroma	0.10 ±0.31	0.56 * ±0.73	0.00
Middle stroma	0.00	0.33 ±0.50	0.00
Posterior stroma	0.30 ±0.47	0.56 ** ±0.53	0.00
Descemet’s membrane	0.05 ±0.22	0.56 * ±0.73	0.00
Endothelium	0.71 *** ±0.69	1.29 ±0.49	1.70 ±0.50

PBK = pseudophakic bullous keratopathy, FECD = Fuchs’ endothelial corneal dystrophy. Numbers indicate semiquantitative scoring ± SD (0 means no staining, 0.5+ minimal, 1+ mild, 2+ moderate, 3+ marked positivity). * = level of significance (*p* ˂ 0.05 *; *p* ˂ 0.01 **; *p* ˂ 0.001 ***).

**Table 6 jcm-11-05991-t006:** Comparison of matrilin-2 expression in the different corneal layers in normal and diseased groups.

	PBK vs. Control	FECD vs. Control	PBK vs. FECD
Epithelium	0.0012	0.4415	0.0021
Bowman’s membrane	0.7336	0.0975	0.1333
Anterior stroma	0.5269	0.0260	0.0391
Middle stroma	0.9999	0.0737	0.0230
Posterior stroma	0.0658	0.0081	0.2371
Descemet’s membrane	0.9999	0.0260	0.0180
Endothelium	0.0009	0.3348	0.0340

PBK = pseudophakic bullous keratopathy, FECD = Fuchs’ endothelial corneal dystrophy. Numbers indicate adjusted *p* values, and red numbers are statistically significant data.

**Table 7 jcm-11-05991-t007:** Aggrecan immunopositivity in the different corneal layers in normal and pathologic conditions.

	PBK	FECD	Control
Epithelium	1.47 ±0.62	1.33 ±0.58	1.3 ±0.82
Bowman’s membrane	1.25 ±1.02	1.66 ±1.22	1.00 ±0.94
Anterior stroma	0.55 * ±0.76	0.44 ±0.73	0.00
Middle stroma	0.8 ±0.95	0.55 ±0.88	0.00
Posterior stroma	1.2 * ±1.32	1.55 ** ±1.33	0.00
Descemet’s membrane	0.00	0.00	0.00
Endothelium	1.14 ±0.36	1.00 ±0.00	0.88 ±0.33

PBK = pseudophakic bullous keratopathy, FECD = Fuchs’ endothelial corneal dystrophy. Numbers indicate semiquantitative scoring ± SD (0 means no staining, 0.5+ minimal, 1+ mild, 2+ moderate, 3+ marked positivity). * = level of significance (*p* ˂ 0.05 *; *p* ˂ 0.01 **).

**Table 8 jcm-11-05991-t008:** Comparison of aggrecan expression in the different corneal layers in normal and diseased groups.

	PBK vs. Control	FECD vs. Control	PBK vs. FECD
Epithelium	0.7146	0.9999	0.9421
Bowman’s membrane	0.5963	0.2199	0.3767
Anterior stroma	0.0296	0.0867	0.6397
Middle stroma	0.0637	0.2105	0.6749
Posterior stroma	0.0113	0.0031	0.5033
Descemet’s membrane	-	-	-
Endothelium	0.2490	0.9999	0.5579

PBK = pseudophakic bullous keratopathy, FECD = Fuchs’ endothelial corneal dystrophy. Numbers indicate adjusted *p* values, and red numbers are statistically significant data.

**Table 9 jcm-11-05991-t009:** Comparison of cumulative immunopositivity of corneal layers in control and diseased groups regarding to tenascin-C, matrilin-2 and aggrecan expression.

	PBK vs. Control	FECD vs. Control	PBK vs. FECD
tenascinC	<0.0001	<0.0001	0.0491
matrilin-2	0.7426	0.0196	0.0028
mggrecan	0.0024	0.0324	0.7530

PBK = pseudophakic bullous keratopathy, FECD = Fuchs’ endothelial corneal dystrophy. Numbers indicate adjusted *p* values, and red numbers are statistically significant data.

## Data Availability

The datasets used and/or analysed during the current study are available from the corresponding author upon reasonable request.

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
