# Peer review of "Expression Pattern of Tenascin-C, Matrilin-2, and Aggrecan in Diseases Affecting the Corneal Endothelium"

_jcm, 2022, doi:10.3390/jcm11205991_

Round 1
Reviewer 1 Report (Previous Reviewer 2)
The paper was revised well. All my concerns were answered. I don't have any objections to the publication.
Author Response
The Authors are thankful for the Reviewer for the constructive criticism throughout the manuscript evaluation process.

Reviewer 2 Report (Previous Reviewer 1)
As they improved the point by point that reviewers suggested, the manuscript had enough information and sophisticated sentences to publish.
Figures also looks nicer compared to last one. I agree the manuscript to be published.
Author Response
The Authors are thankful for the Reviewer for the constructive criticism throughout the manuscript evaluation process.

This manuscript is a resubmission of an earlier submission. The following is a list of the peer review reports and author responses from that submission.
Round 1
Reviewer 1 Report
For authors
The study was impressive, and authors message is very simple and important. I think the manuscript should be read for the corneal surgeons who are doing corneal transplantation.
I have a few comments as minor revision.
The point by point should be addressed; below
1. Figure 6: In Matrilin-2 immunohistochemistry part, authors described corneal specimens B is PBK, C is FECD. The staining of specimen B looks more moderate compared to specimen C. But authors conclusion is opposite. And the setting of the background (white color) is also different. Authors should replace the picture with nicer picture in Figure 6 part.
2. In discussion part, as authors described tenascin-C production levels reflects infection, inflammation, wound-healing processes. And the authors results showed significant tenascin-C positivity in PBK compared to FECD. I thought there are difference between early phase PBK and late phase PBK. Late phase PBK has more accumulation of inflammation and fibroblasts. My suggestion is to show information the background of PBK patients precisely. Which patients did authors select in PBK criteria.
Reviewer 2 Report
The authors showed Tenascin-c, matrilin-2 and aggrecan immunostaining in healthy, PBK and FECD human corneas. The diseased corneas have higher expressions of these proteins compared to the healthy ones. Some of these finding in human corneas were reported by the same group and others, but this study still has updated and new insights.
However, there are several major concerns. Some of them are critical and require major revision.
1) The authors did not perform the semiquantitative analysis following the reference 25. Current analysis (scoring) is not semiquantitative but qualitative. If you want to keep “semiquantitative”, please conduct new analysis following the ref 25. However, the counter staining often interferes the analysis. It is fine to keep current qualitative scoring. In that case, please fix the manuscript.
2) The immunohistochemistry (line 147-159) was not described well. The protocol of MACH 4 Universal HRP-Polymer indicated the procedure is different between mouse and rabbit antibody. Please describe what you did exactly. Also, for the matrilin-2, the author used alkaline phosphatase detection system. Please write the details.
3) For the immunohistochemistry, the control is necessary. Isotype controls are preferred. Non-primary control is not recommended but acceptable. The negative controls should be performed side by side, but it is impossible now. Please perform some negative controls in the healthy control sections. Because the diseased samples are precious and limited, the negative control in the diseased corneas is not required.
4) The demographic of the healthy cornea was not described. Especially, the diseased corneas are female-dominant. How about the healthy corneas? If available, please mention it. Also, I think the diseased corneas were fixed immediately after harvesting. How about the healthy corneas? Is the time between death and preservation available? Did you use any corneal preservative medium such as OptiSol? The protein expression may change depending on the time. Even if some parameters are different, it will not lose the significancy of paper, but it is important factor for the readers. If not available or lost, please say so in the manuscript.
5) Statistics (Line 176-183). At first, the Shapiro-Wilk normality test is not required in this case. The results of Kruskal-Wallis test were not shown. It was unclear which test (Mann-whitney test or Dunn’s test) is used as a post-hoc analysis in this paper. In the case of Mann-whitney test, did you use Bonferroni correction? Even if not, it is OK. Please describe them correctly.
6) Current figure 1 should be the last as summary of results. The most important figures are immunostaining images. The readers want to see the immunostaining images rather than the tables of p-values. Also, some tables and figures showed the same things. One of them are enough. I strongly encourage the authors to reconstruct the manuscript.
7) The immunostaining images did not correspond to the average of the scoring data. For example, in figure 6, strong immunostaining was observed in the epithelium of FECD sample, which intensity was similar to the PBK sample. However, the figure 3 showed the immunostaining score in the epithelium of FECD samples was much lower than the PBK’s one. Usually, the representative images should be corresponding to the graph. Like this, I found many inconsistences between the images and graphs.
8) The comparison among the different layers like table 5 is really not interesting. If the authors want to keep them, it is fine. But I think more immunostaining images and more examples would be more helpful and useful for the researchers.
9) Also, do you need the figure8 and 9? I don’t think so. However, if you want to keep them, please add “Proposed” or “hypothesized” in the front of the figure titles, e.g. “Proposed wound healing mechanisms in PBK”.
10) Usually, I do not deny the discussion. But this discussion is just too long and not interesting, since the hypothesis is on the hypothesis again. Many ideas are beyond the current results.
11) The title is like a review paper. Please reconsider the title.